# The Kinematic and Kinetic Responses of the Trunk and Lower Extremity Joints during Walking with and without the Spinal Orthosis

**DOI:** 10.3390/ijerph19116952

**Published:** 2022-06-06

**Authors:** Chenyan Wang, Xiaona Li, Yuan Guo, Weijin Du, Hongmei Guo, Weiyi Chen

**Affiliations:** College of Biomedical Engineering, Taiyuan University of Technology, Taiyuan 030024, China; wangccccyyy@163.com (C.W.); lixiaona@tyut.edu.cn (X.L.); guoyuan@tyut.edu.cn (Y.G.); duweijin0156@link.tyut.edu.cn (W.D.); guohongmei@tyut.edu.cn (H.G.)

**Keywords:** gait, poor posture, spinal orthosis, musculoskeletal modeling

## Abstract

Spinal orthoses are an effective option for restoring the spine to its original position and controlling poor posture. However, the effects of poor posture and spinal orthoses on the kinematics and kinetics of trunk and lower extremity joints remain unclear. A six-camera Vicon motion capture system and two AMTI force plates were employed to collect gait parameters, including joint angle (spine, thorax, hip, knee, and ankle), range of motion (ROM), and ground reaction forces (GRFs). Furthermore, joint moments and joint reaction forces (JRFs) were calculated using a full-body musculoskeletal model in OpenSim. One-way repeated-measures ANOVA (*p* < 0.05) was used to compare significant differences among three trial conditions. These three conditions were walking in a normal posture, poor posture, and spinal orthosis. The results showed that spine ROM in the coronal and transverse plane was significantly lower when walking with a spinal orthosis compared to walking in normal and poor posture (*p* < 0.05). Compared to normal posture, the lumbar moments and back compressive forces were significantly increased when walking in poor posture (*p* < 0.05). However, when walking with a spinal orthosis, there was a significant decrease in trunk moments and reaction forces compared to walking in poor posture (*p* < 0.05). Individuals with poor posture could potentially induce instability and disorders, as evidenced by an increase in trunk moments and JRF compared to the normal posture. Spinal orthosis not only restricts spine ROM but also reduces the load on the spine and thus increases balance and stability.

## 1. Introduction

### 1.1. Background

Body posture shows the interrelationship of human muscles, nervous system, bones, and internal organs [1,2]. Normal posture refers to the head in a normal position (not leaning forward, backward, laterally, or twisted), the cervical vertebrae, thoracic vertebrae, and lumbar vertebrae are in normal curvature, the pelvis and hip joints are in a normal position (not abducted or adducted), and the ilium and pubic symphysis are in the same plane [3]. The body posture that deviates from the normal state is known as poor posture and is one of the problems associated with physical development in children [4]. Poor posture mainly includes trunk lean, forward head posture, anterior pelvic tilt, postural kyphosis, and knee hyperextension [5,6,7]. With the rapid development of digital technology, the use of intelligent electronic devices is very common in many countries and areas [8]. However, prolonged use of smart devices is associated with an increased risk of poor posture. Between 22 and 65% of children and adolescents exhibit poor posture [9]. Adolescents with poor posture have a higher frequency of functional and structural pathologies, such as headaches, lumbar pain, and soft tissue damage. Moreover, poor posture in adolescence may also lead to the aforementioned problems in adulthood [10]. Over time, these can cause imbalance and instability and increase the risk of falls in older adults [11].

From the viewpoint of both morphological evolution and functional adaptation, the spinal function is related to normal morphology, and natural spinal alignment could allow for full biomechanical advantage in humans [12,13]. Immobilization or dynamic postural correction by applying corrective forces through spinal casts or braces is an important approach to reducing poor posture [14]. Spinal orthoses have been designed to reduce trunk range of motion [15,16], improve trunk stiffness, increase spinal stability [17], and correct deformities [18].

### 1.2. Related Work

There were many studies that have explored the function of the spinal orthosis. Shahvarpour et al. [19] evaluated the biomechanical outcomes of participants wearing orthosis while performing daily tasks. They measured the angular kinematics of the trunk and thigh by using a 3D inertial motion system and found that orthosis could provide motion restriction. Namdar et al. [20] assessed the efficacy of back orthosis and posture training support on walking ability using the mobility scale test, 2-min walk test, and 10-m walk test and concluded that both orthosis and posture training support played an effective role in improving walking function. However, only a few studies have reported gait biomechanical results when walking with a spinal orthosis, leaving the impact of spinal orthoses on the overall kinematics and kinetics incomplete.

With rapid developments in recent years, three-dimensional gait motion capture analysis has been performed for the detailed analysis of participants with healthy [21] and various spinal deformities [22]. These motion capture systems could record and analyze the kinematics of the human body in the sagittal, frontal, and transverse planes [23]. Alijanpour et al. captured kinematic data through a seven-camera motion capture system and compared the spine-pelvis coordination and coordination variability between rowers with and without low back pain [24]. Shiba et al. [25] demonstrated significant differences in sagittal alignment between dynamic and static parameters in patients with degenerative lumbar kyphoscoliosis through gait analysis. Similarly, Haddas et al. [26] evaluated kinematics, ground reaction forces, and electromyography results by gait analysis and found significant changes in the spine and lower extremity values in the abnormal spinal sagittal alignment. 

Although many factors may contribute to back disorders, peak moments and joint reaction forces during movement have been identified as strong correlates [27]. Direct measurement of these parameters requires invasive procedures, such as implanted sensors, which can cause damage to the human body. Musculoskeletal models provide an alternative way to evaluate biomechanical risk factors [28]. Sasaki et al. [29] conducted a gait motion analysis combining a three-dimensional musculoskeletal model produced from whole-body computed tomography and magnetic resonance imaging and evaluated the sagittal alignment of the spine and lower extremity joints during standing and walking. They found that the subject-specific musculoskeletal model could provide greater insight into the spine, pelvis, and leg. Molinaro et al. [30] also computed the lumbar moments and peak joint reaction forces when throwing bags of different weights using a full-body musculoskeletal model in OpenSim.

Previous studies have examined the effects of poor posture and spinal orthosis on daily life. However, only a few studies have reported gait biomechanical results of the trunk and lower extremity joints when walking in poor posture and with a spinal orthosis. There is an urgent need to investigate three-dimensional kinematics and kinetics of the trunk and lower extremity joints when walking in poor posture and with a spinal orthosis. Thus, the first purpose of the current study was to assess the effects of abnormal spinal sagittal alignment (poor posture) on the trunk and lower extremity joint angles, moments, and reaction forces during walking. The second purpose was to compare the joint kinematics and kinetics of the three gait patterns, including walking in a normal posture, poor posture, and wearing a spinal orthosis, thus further exploring the function of the spinal orthosis. We hypothesized that during walking, people in poor posture would exhibit greater moments and joint reaction forces (JRFs) compared to normal posture. Moreover, there was no significant difference between these parameters when people walked in the normal posture and with a spinal orthosis.

## 2. Methods

### 2.1. Participants

A total of twelve participants (6 females and 6 males: age 23.4 ± 3.0 years, height 164.5 ± 9.0 cm, weight 59.6 ± 8.6 kg, and BMI 21.9 ± 1.94 kg/m^2^) were recruited for this experiment. All participants had no lower extremity injuries within the previous six months, no joint instability, or any neurological condition that could potentially affect gait. Each participant was informed about the experiment and signed written consent. Ethical approval was obtained from the University Research Ethics Committee (TYUT202105003). The procedures used in this study adhere to the tenets of the Declaration of Helsinki.

### 2.2. Measurements

Three-dimensional body movements during gait were collected in this study. Kinematic data were recorded at 100 Hz using the Vicon motion capture system (Vicon, Oxford, UK), which includes six infrared cameras. Two AMTI force plates were embedded in the laboratory floor to collect ground reaction forces (GRFs) and detect the gait cycle at a frequency of 1000 Hz. The Vicon Plug-in Gait Full Body set of markers (39 light-reflecting markers with 14 mm diameter) was placed on the anatomical skeleton landmarks of each participant, which provided a comprehensive view of segmental motion [31]. Anthropometric measurements (shoulder offset, elbow width, wrist width, hand thickness, leg length, knee width, and ankle width) were taken before building a Full Body model in the Vicon system. Female participants wore a sports bra and black shorts, while male participants wore black shorts and no shirt. Moreover, all participants wore athletic socks and no shoes. A spinal orthosis was selected to evaluate the efficacy of joint kinematics and kinetics. Details of markers locations and the spinal orthosis are displayed in Figure 1. The back support structure of the spinal orthosis (CO-29, Ober, China) consists of a 3 mm thick titanium plate whose width varies with its length. In the length range of 0–35 cm, the width is 8.5 cm, while in the length of 35–47 cm, the width is 5 cm. The participant wore the spinal orthosis and then was adjusted by the same experienced researcher.

Three gait patterns, including walking in a normal posture, poor posture, and wearing a spinal orthosis, were selected and measured to explore their effects on joint kinematics and kinetics. In addition, poor posture was achieved by participants leaning their trunks forward, in which the spinal angle was obviously changed in the sagittal plane. Participants were asked to stand in a normal posture, looking straight ahead, with their arms hanging naturally (Figure 2a). The participants were then instructed to modify their trunk angle according to a posture assessment analysis chart placed at their side (Figure 2b). Kinematic data (spine, thorax, hip, knee, and ankle angles) were calculated from the trajectories of light-reflecting markers in the Vicon Nexus 2.6 software (Vicon, Oxford, UK).

Participants were instructed to walk at a self-selected comfortable walking speed. During the per walking test, the gait cycle was defined as consecutive heel strikes of the ipsilateral foot [32]. Six acceptable trials were completed for each subject.

### 2.3. Musculoskeletal Analysis

The musculoskeletal analysis has been completed in OpenSim [30,33], which is used to analyze the joint inverse dynamics (ID) and JRFs (Figure 3). The generic musculoskeletal model Gait2392_Simbody, which consists of 10 rigid bodies, 92 musculotendon actuators, and 23 degrees of freedom [34,35], was applied in this analysis. Static data for normal posture and poor posture (trunk lean) were collected by the Vicon motion capture system. Furthermore, the personalized musculoskeletal model (normal and poor posture) for each participant was obtained by inputting anthropometric measurements and the location of 39 light-reflecting markers in the Scaling Tool [30]. Then, joint angles that match the experimental motion data were obtained in the inverse kinematics (IK) tool [36]. The IK results were important for the accuracy of the kinetic results (moments and JRFs) calculated using ID, static optimization (SO), and JRF analysis. ID was applied to calculate joint moments to correct the torque distribution under the coupler constraints of the model [37]. The moments of the lumbar, hip, knee, and ankle joints in the sagittal plane and the moments of the lumbar and hip joints in the coronal plane were then calculated in this study. SO was used to estimate muscle forces and muscle activations that satisfy a given movement and GRFs [38]. Moreover, compressive reaction forces were also calculated for the back, hip, knee, and ankle joints using the JRF analysis in OpenSim [39].

### 2.4. Data Analysis

The kinematic data (joint angles) were obtained in the Vicon system. The raw marker trajectories and GRFs were stored in .c3d file formats and converted to OpenSim file formats (.trc and .mot files) by Matlab software [40]. Moreover, these data were low-pass filtered, which used a zero-lag 2nd order Butterworth filter with cut-off frequencies of 6 and 10 Hz, respectively [41]. Moments and JRFs were then calculated in the OpenSim software using ID, SO, and JRF analysis [42]. Joint angles, moments, and JRFs were time-normalized to 100% of the gait cycle [31]. In addition, joint moments were normalized relative to body weight × height (BW × H) % [35], and JRFs were normalized relative to body weight (BW) [36,43]. Moreover, the minimum value of each gait was subtracted from the maximum value to determine the ROM of the joints [24].

One-way repeated measures of variance (ANOVA) were applied to determine the significance of kinematic and kinetic variables among different gait patterns. Post hoc Bonferroni adjustment analysis was applied to compare statistical significance for each condition [31]. All statistical analyses were conducted using SPSS 21 (SPSS Inc., Chicago, IL, USA), and the significance level was set to 0.05.

## 3. Results

### 3.1. Joint Kinematics

The trunk and lower extremity joint angles for three gait conditions are shown in Figure 4. When walking in poor posture, the spine and thorax exhibited greater flexion angles in the sagittal plane during the whole gait cycle. Moreover, the axial rotation angles of the thorax were greater in the terminal stance, pre-swing, initial swing, and mid-swing (30–80% of the gait cycle) when walking in poor posture compared to walking in normal posture and spinal orthosis. When walking with a spinal orthosis, the lateral bending and axial rotation angles of the spine during the gait cycle were lower in both the coronal and transverse planes. At the hip, knee, and ankle joints, there were no obvious differences in sagittal plane (hip flexion, knee flexion, and ankle plantarflexion) movement.

### 3.2. ROM of Joints

The ROM of the joints in three groups is shown in Table 1. The spine ROM in the coronal and transverse plane was significantly lower when walking with a spinal orthosis compared to walking in normal and poor posture (*p* < 0.05). Moreover, when walking with a spinal orthosis, the ROM of the thorax in the coronal plane was significantly lower than walking in poor posture (*p* < 0.05). In addition, when walking with a spinal orthosis, the hip showed significantly higher ROM compared to walking in poor posture (*p* < 0.05). No significant differences were observed in the ROM of any joint when walking in normal and poor posture (*p* > 0.05).

### 3.3. Joint Moments

Joint moments computed using the ID tool are presented in Figure 5, and statistical analysis of maximum values is shown in Table 2. When walking in poor posture, the lumbar moments exhibited greater values in the sagittal plane during the whole gait cycle and in the coronal plane during early stance and early swing (0–20% and 60–80% of the gait cycle). Compared to normal posture, the maximum lumbar flexion and extension moments were significantly larger when walking in poor posture (*p* < 0.05). However, no significant differences were found in lumbar flexion and extension moments when walking in normal posture and spinal orthosis (*p* > 0.05). The results also showed no statistically significant differences in hip adduction/abduction, knee flexion/extension, and ankle plantarflexion/dorsiflexion moments when walking in a normal posture, poor posture, and spinal orthosis (*p* > 0.05).

### 3.4. Joint Reaction Forces

The compressive force of the back, hip, knee, and ankle joints are presented in Figure 6, and statistical analysis of the maximum values is listed in Table 3. When walking in poor posture, the greater back compressive force was exhibited during the whole gait cycle. However, the trends and values of back compressive forces were similar when walking in normal posture and spinal orthosis. Additionally, at the hip, knee, and ankle joints, there was no significant difference in the maximum compressive forces when walking in a normal posture, poor posture, and spinal orthosis (*p* > 0.05).

## 4. Discussion

This study provides a comparison of kinematics and kinetics of the trunk and lower extremity joints when walking in a normal posture, poor posture, and spinal orthosis. The results showed that joint kinematics and kinetics were indeed influenced when walking in poor posture, especially for spine kinematics, lumbar moments, and back compressive forces. The present study also indicated that the kinematics and kinetics of the trunk were altered when participants walked with a spinal orthosis.

The development of sagittal posture is important to achieve the normal physiological curve [44]. In addition, a normal physiological curve is crucial for the chest volume and lung ventilation function [45]. However, poor posture occurs in almost every field and is strongly associated with functional loss and physical disability [2]. The measurement of joint kinematics plays an important role in injury and performance investigations [46]. In this study, the effects of spinal morphology changes induced by trunk forward lean (poor posture) on joint kinematics were quantitatively evaluated by 3D gait motion analysis. Spine angles have been employed in many biomechanical and ergonomic studies, often associated with long-term static or dynamic postures [47]. The kinematic results showed an increase in spine angles when walking in poor posture. These increased spine angles may cause the center of mass (COM) to move forward and induce the COM to be outside the stability limits in the standing posture, which leads to imbalance [48]. However, there were no significant differences in the hip, knee, and ankle flexion/extension angles when walking in normal and poor posture, which suggested that poor posture has little effect on lower extremity joints.

Poor posture can be improved by training to strengthen the back muscles or by the spinal orthosis to apply corrective forces and remind the wearer to maintain normal posture [9]. Spinal orthoses have been designed to prevent curve progression and reduce deformities of the spine [14]. Previous studies have demonstrated that wearing a spinal orthosis can be useful in improving posture. Azadinia et al. [48] discovered that spinal orthoses could help correct anteversion and reduce kyphosis angle through a prospective randomized study. Similarly, compared to walking in normal or poor posture, the kinematic results demonstrated a reduction in spine and thorax flexion angles during the whole gait cycle when walking with a spinal orthosis. The results showed that spine ROM was significantly lower in the coronal and transverse planes when walking with a spinal orthosis. This is in line with the previous findings by Zhang et al., who found that the spinal orthosis could provide a restriction on ROM in trunk movements [49]. Moreover, there was no significant difference in ROM of the lower extremity joints (hip, knee, and ankle) when walking in normal posture and a spinal orthosis, and it indicated that spinal orthosis has little effect on the lower extremity joints.

Furthermore, moments and JRFs of the trunk and lower extremity joints were also evaluated when walking in poor posture. Compared to normal posture, lumbar flexion moments and back compressive forces are greater when walking in poor posture, which is consistent with the point that postural changes may put additional physical stress on the bones [50]. The results showed that the max back JRF when walking in a poor posture was about 1.75 times higher than the normal posture. There is a relationship between back loading and the development of back disorders, such as low back pain [51]. Considering that low back pain is the result of cumulative injury in the lower back [52], it is recommended to maintain a normal posture when walking to prevent back damage. Moreover, participants with trunk forward lean (poor posture) required higher muscle activation compared to normal posture. The results also indicated that angle changes in the sagittal plane bring a greater load to the trunk. Therefore, when people maintain a poor posture for a long time, it can lead to muscle fatigue, spinal imbalance, and eventually pathological changes [53]. Moreover, there was no significant difference in the moments and compressive forces of the lower extremity joints (hip, knee, and ankle) when walking in normal or poor posture, demonstrating that poor posture has relatively little effect on the kinetics of the lower extremity joints.

Regarding the kinetic parameters when walking with a spinal orthosis, the moments at the lumbar, hip, knee, and ankle joints were calculated in this study. Compared to walking in poor posture, there was a statistically significant reduction in the maximum flexion/extension lumbar moment when walking with a spinal orthosis. The use of the spinal orthoses did not significantly affect the lumbar moment compared to walking in normal posture; however, a slight decrease in the lumbar moment could still be observed. Furthermore, the static optimization method predicted a decrease of 41.8% in maximum back compressive force when walking with a spinal orthosis, respectively, compared to walking in poor posture. Long-time abnormal weight bearing on the lumbar and back may increase the risk of acute or chronic pain [54]. Therefore, reducing abnormal stress on the lumbar and back is key to relieving pain and improving quality of life. The results of this study showed that the use of spinal orthosis significantly reduced stress on the lumbar and back and indicated that spinal orthosis is an effective way to improve poor posture.

There are still a few shortcomings in the present study that should be considered. First, this study focused only on the immediate effects when walking with the spinal orthosis and ignored gender differences. Although a certain familiarization period was allowed, it is possible that participants would walk differently with increased time of use. Long-term follow-up studies on these areas are valuable and might reveal further insights. Second, participants walked only at a self-selected speed, while Lerner et al. indicated that joint moments were related to walking speed [55]. Future research should control walking speed when analyzing joint kinematics and kinetics during gait movement. Third, the sample size was relatively small, and only young participants were recruited, which was a preliminary study. In the future, more participants of different age levels will be recruited to make the findings more generalizable. Finally, when participants walked with a spinal orthosis, the musculoskeletal model did not include a spinal orthosis. Although comparative studies of kinetic results in participants walking with the spinal orthosis are possible, the addition of the spinal orthosis to the musculoskeletal model could further improve the accuracy of results in the future study.

## 5. Conclusions

The current study expands the knowledge of the previous finding by reporting not only the 3D gait motion when walking in poor posture but also the trunk and lower extremity kinematics and kinetics when walking with a spinal orthosis. There was a significant increase in spine and thorax angle when walking in poor posture compared to walking in a normal posture, while walking with a spinal orthosis showed a similar trend and values. This result indicated that the spinal orthosis could restrict the trunk motion and prevent poor posture. Furthermore, when walking in poor posture, greater lumbar moments and back compressive forces were exhibited, and prolonged abnormal weight bearing may affect the quality of life. However, lumbar moments and back compressive forces were significantly decreased when walking with spinal orthosis, suggesting that it can be an effective way to improve poor posture. Moreover, results for the lower extremity joints showed that there was no significant difference in the kinematics and kinetics at the hip, knee, and ankle joints when walking in normal posture and spinal orthosis.

## Figures and Tables

**Figure 1 ijerph-19-06952-f001:**
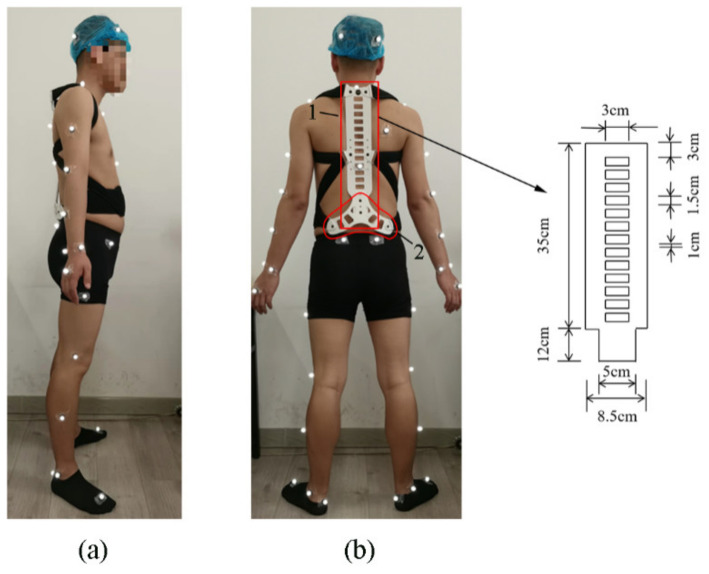
Reflective markers were attached to the anatomical skeleton landmarks of a subject. (**a**) Side view of the spinal orthosis; (**b**) back view of the spinal orthosis (1 titanium alloy plate and 2 plastic plates).

**Figure 2 ijerph-19-06952-f002:**
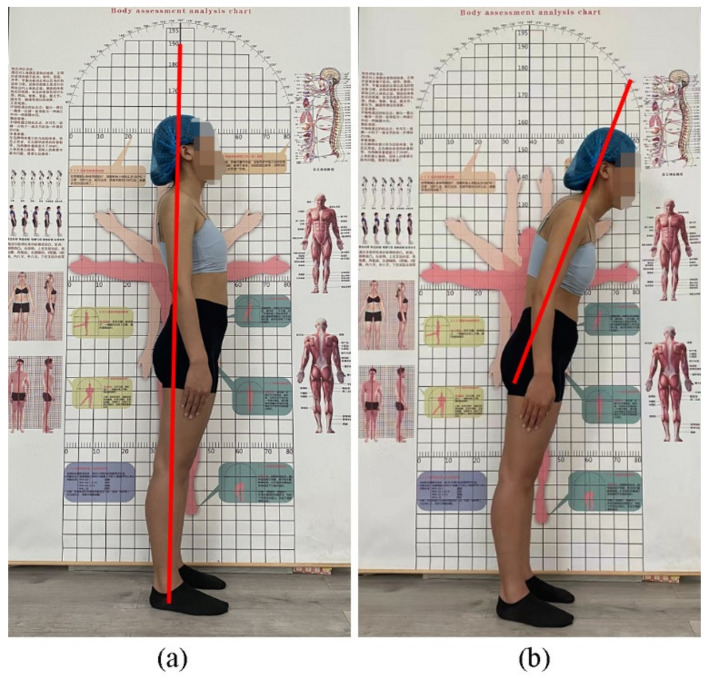
Posture assessment analysis. (**a**) Normal posture; (**b**) poor posture.

**Figure 3 ijerph-19-06952-f003:**
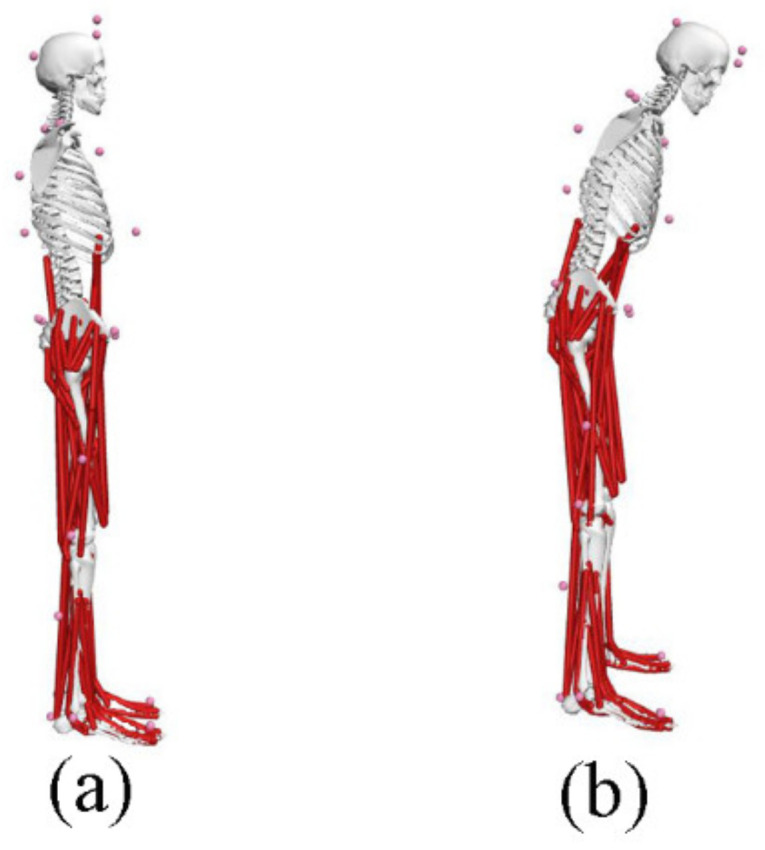
Three-dimensional musculoskeletal model. (**a**) Normal posture; (**b**) poor posture.

**Figure 4 ijerph-19-06952-f004:**
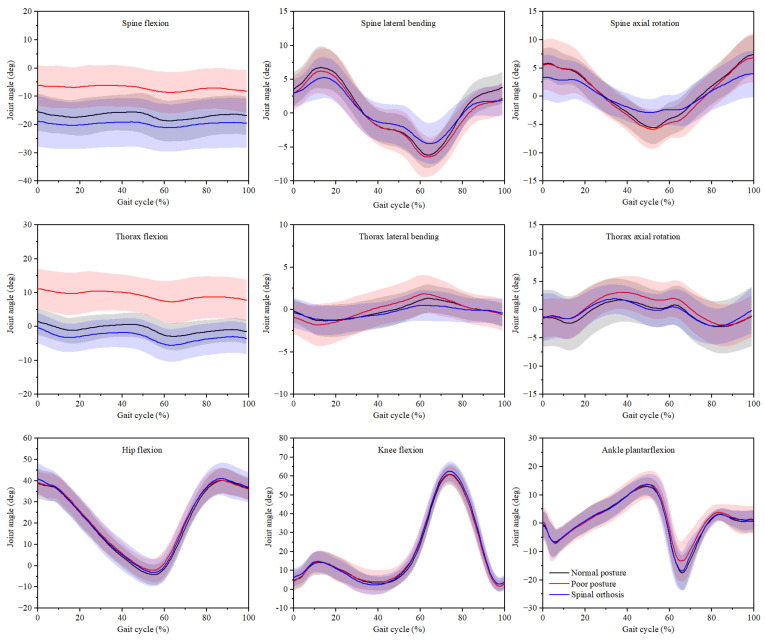
Joint angles (mean and standard deviation) of the trunk and lower extremity joints during the gait cycle when walking in normal posture, poor posture, and spinal orthosis.

**Figure 5 ijerph-19-06952-f005:**
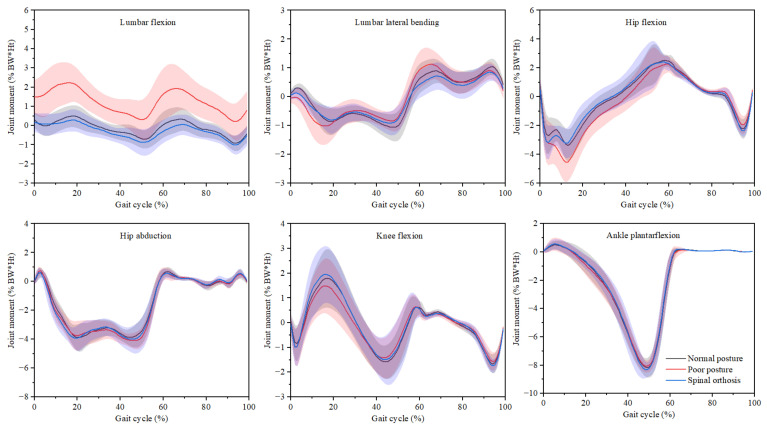
Joint moments (mean and standard deviation) during the gait cycle when walking in normal posture, poor posture, and spinal orthosis.

**Figure 6 ijerph-19-06952-f006:**
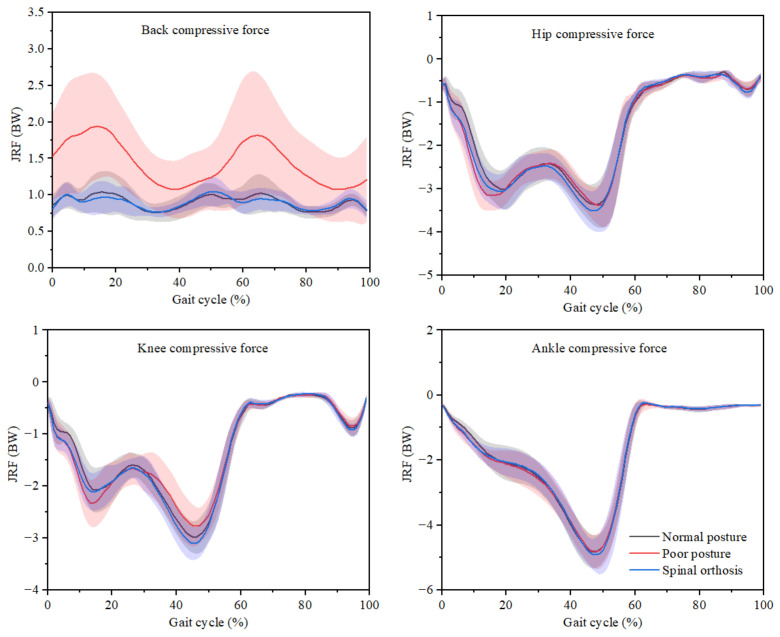
Joint reaction forces (mean and standard deviation) during the gait cycle when walking in normal posture, poor posture, and spinal orthosis.

**Table 1 ijerph-19-06952-t001:** ROM of the joints during the gait cycle (mean ± SD).

Joints ROM (Degree)		Normal Posture	Poor Posture	Spinal Orthosis	*p*
Spine	Sagittal plane	4.35 (1.60)	3.90 (1.57)	3.08 (1.81)	0.06
	Coronal plane	13.41 (3.27)	13.14 (4.29)	10.18 (3.92) *^#^	<0.01
	Transverse plane	13.04 (3.16)	13.24 (5.33)	7.26 (3.20) *^#^	<0.01
Thorax	Sagittal plane	4.64 (1.45)	4.83 (2.01)	5.29 (1.48)	0.42
	Coronal plane	3.38 (1.72)	4.10 (2.29)	2.63 (1.42) ^#^	0.03
	Transverse plane	7.32 (1.83)	7.15 (1.99)	6.97 (2.30)	0.83
Hip	Sagittal plane	45.13 (2.97)	43.07 (4.03)	45.63 (3.45) ^#^	<0.01
Knee	Sagittal plane	59.43 (2.96)	60.71 (3.53)	62.40 (2.26)	0.09
Ankle	Sagittal plane	31.75 (2.02)	29.52 (1.99)	32.74 (2.21)	0.11

* Significantly different from normal posture; ^#^ significantly different from poor posture. SD—standard deviation.

**Table 2 ijerph-19-06952-t002:** Statistical analysis of the maximum joint moments during the gait cycle (mean ± SD).

Joint Moment (%BW × H)	Normal Posture	Poor Posture	Spinal Orthosis	*p*
Max lumbar flexion	0.63 (0.59)	2.49 (1.06) *	0.46 (0.55) ^#^	<0.05
Max lumbar extension	1.06 (0.42)	−0.06 (0.98) *	1.10 (0.48) ^#^	<0.05
Max lumbar left LB	1.15 (0.29)	1.27 (0.49)	0.99 (0.35)	0.13
Max lumbar right LB	1.14 (0.43)	1.28 (0.52)	1.18 (0.45)	0.33
Max hip flexion	3.00 (0.53)	2.81 (0.67)	3.08 (0.75)	0.43
Max hip extension	3.93 (1.13)	4.65 (1.39)	3.64 (0.94) ^#^	<0.05
Max hip adduction	1.00 (0.23)	0.99 (0.23)	0.96 (0.20)	0.80
Max hip abduction	4.52 (0.68)	4.37 (0.51)	4.49 (0.67)	0.62
Max knee flexion	1.96 (1.03)	1.81 (0.87)	2.13 (1.09)	0.13
Max knee extension	2.14 (0.33)	2.00 (0.55)	2.17 (0.51)	0.44
Max ankle plantarflexion	0.57 (0.25)	0.67 (0.32)	0.64 (0.31)	0.24
Max ankle dorsiflexion	8.41 (0.72)	8.16 (0.43)	8.53 (0.61) ^#^	0.03

* Significantly different from normal posture; ^#^ significantly different from poor posture. SD—standard deviation; LB—lateral bending. Joint moments were normalized relative to body weight × height (BW × H)%.

**Table 3 ijerph-19-06952-t003:** Statistical analysis of the maximum JRFs during the gait cycle (mean ± SD).

JRF (BW)	Normal Posture	Poor Posture	Spinal Orthosis	*p*
Max back JRF	1.19 (0.22)	2.08 (0.76) *	1.21 (0.13) ^#^	<0.05
Max hip JRF	3.66 (0.49)	3.64 (0.35)	3.69 (0.46)	0.85
Max knee JRF	3.04 (0.26)	2.97 (0.32)	3.17 (0.30)	0.09
Max ankle JRF	4.96 (0.52)	4.88 (0.52)	5.05 (0.58)	0.23

* Significantly different from normal posture; ^#^ significantly different from poor posture. SD—standard deviation; JRF—joint reaction force. JRFs were normalized relative to body weight (BW).

## Data Availability

The original contributions presented in the study are included in the article. Further inquiries can be directed to the corresponding authors.

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
