# Peer review of "The Kinematic and Kinetic Responses of the Trunk and Lower Extremity Joints during Walking with and without the Spinal Orthosis"

_ijerph, 2022, doi:10.3390/ijerph19116952_

Round 1
Reviewer 1 Report
The authors investigated the effects of posture and orthosis for joint angle, range of motion, joints moment and ground reaction forces during gait. Their results are supported by their experiment results. The reviewer just has several questions:
- The authors investigated the effects of orthosis for gait parameters. However, did the subjects in this study have spinal deformities? It is doubtful that the orthosis would have any effect on subjects without a spinal deformity. Is the gait pattern with orthosis different from the gait pattern of normal posture?
- I did not understand the use of posture assessment analysis chart used to define poor posture. Could you please explain how this chart was used?
- In this study, a significant difference of about 0.8 BW in the mean value of max back JRF was observed in the poor posture compared to the normal posture. However, it is unclear how clinically significant this difference is. I suggest to add to the discussion whether this difference is clinically significant or not.
- Line 86-93: In this part, the purpose of this study was stated. How about adding a hypothesis to this paragraph? It would be more helpful in interpreting the results if you could describe how you hypothesize that posture affects gait based on previous research.
Author Response
Thank you for your comments on our manuscript, the point-by-point responses can be viewed in the attachment.
Please see the attachment.

Reviewer 2 Report
In this paper, the authors compared kinematic and kinetic responses during 3 different walking conditions. This is an interesting paper, however, some modifications are required. Please refer to the below comments:
1. Authors should review the manuscript to make sure that there are no English/Grammatical errors such as:
Page 1 (Line 37): "There are 22 to 65% of children and adolescents exhibit poor posture:
2. Healthcare costs, this paper was published in 2016. Authors should also cite never work. Also, it is important to consider how new IoT enabled healthcare is affecting the Young vs. Older generations and related costs
"Ultimately, poor posture can increase healthcare costs and 42 lead to a huge economic burden [13]"
3. Authors mentioned that "Pfeifer et al. [20] demonstrated that the use of orthoses could 52 reduce swaying and subsequently improve stability in a randomized trial. Van Poppel et 53 al. [21] also found that spinal orthoses could provide significant motion restriction 54 through a literature review".
These papers were published almost two decades ago (2004 and 2006). There has been lots of research/improvements achieved in the field since then. These papers are inadequate to make this claim. Authors need to cite recent papers and provide details (i.e., shortcomings) to justify how their work adds to the knowledge.
4. Introduction sections need substantial improvement. Consider splitting this section into "Introduction" and "Background work" sections
5. Only 12 participants were recruited in the study (6 female and 6 male) with an average age of 23.4 years. Why didn't the authors recruit more participants and was there any power analysis conducted before recruitment to justify these numbers.
6. Moreover, how can data from these healthy young adults be mapped to the children/elderly population, given that human gait can vary a lot between these 3 age groups (children, Young Adults, and Elderly Population).
7. Authors mentioned that a Butterworth filter with cut-off frequencies of 6 and 10Hz was used. What were the reasons behind this approach? Human gait is generally much slower under spinal issues.
8. Were participants trained on how to walk in bad posture and how was it decided that participants have learned this skill.
9. Did all participants walk in the same order (i.e., good posture, bad posture, and orthosis)?
10. Authors should clearly list the limitations of current work (i.e. only healthy young participants etc) and how they plan to address this in the future.
e only two
Author Response
Thank you for your comments on our manuscript, point-by-point responses can be viewed in the attachment.
Please see the attachment.

Round 2
Reviewer 1 Report
Thank you for your responses.
Reviewer 2 Report
Thank you for addressing my concerns